# From Abstract Noise to Architectural Form: Designing Diffusion Models for Efficient Floor Plan Generation

## Abstract

In contemporary architectural design, the generation of innovative and efficient floor plans remains a critical challenge. This research introduces a novel application of diffusion models, specifically adapted for the generation of architectural floor plans. Unlike traditional generative models that broadly target image generation, our approach harnesses the state-of-the-art in diffusion technology to produce detailed, functional, and visually appealing architectural designs. We demonstrate that diffusion models, when finely tuned and conditioned, not only embrace 'implicit, human-learned' architectural semantics but also enhance design efficiency and creativity. The paper details our methodology from adapting the U-Net architecture within diffusion frameworks to incorporating advanced upscaling techniques, significantly reducing computational overhead while maintaining high-resolution outputs. Our results show a promising direction for integrating AI in architectural design, opening new avenues for automated, creative design processes that could revolutionize the industry.

## 1 Introduction

The fusion of artificial intelligence (AI) with architectural design has opened new pathways for innovation in how spaces are conceived and visualized. Recent advancements in generative models, particularly diffusion models, have shown unprecedented success in image-generation tasks. However, their application in specialized domains like architectural design, where detail, accuracy, and adherence to design principles are paramount, remains largely unexplored. This study seeks to bridge this gap by adapting and enhancing diffusion models for the specific task of generating architectural floor plans.

The motivation behind this research is twofold: firstly, to explore the potential of state-of-the-art AI models to understand and implement complex, implicit rules that govern architectural aesthetics and functionality; and secondly, to provide a tool that significantly augments the architect's ability to generate diverse design alternatives quickly. By focusing on the specific use case of floor plan generation, we aim to demonstrate how diffusion models can be meticulously tailored to not only generate images but to do so in a way that adheres to professional architectural standards.

Our approach involves a customized adaptation of the U-Net architecture configured within a diffusion modeling framework. This adaptation is geared towards capturing the nuanced requirements of architectural designs, including the layout of spaces and their functional relationships. Further, we employ upscaling techniques post-generation, allowing the model to operate efficiently at lower resolutions without sacrificing output quality, thus addressing the dual challenges of detail fidelity and computational efficiency.

This paper outlines our comprehensive methodology, from dataset preparation and model architecture design to the detailed training regimen and the subsequent image enhancement techniques. We present empirical results that illustrate the model's capability to produce professional-grade floor plans and discuss the potential applications of this technology in real-world architectural practices. Lastly, we explore future enhancements that could enable more interactive and user-specific design capabilities, underscoring the transformative potential of AI in the architectural field.

## 2 DATASET

### 2.1 RATIONALE FOR DATASET SELECTION

This dataset was specifically chosen for its comprehensive and multi-faceted representation of architectural floor plans, which is crucial for training our diffusion model to recognize and generate realistic and functionally coherent designs. The inclusion of distinct images for walls, room segmentation, and the overall floor plan allows the model to learn and reproduce the structural integrity and the functional zoning of architectural spaces. Additionally, the descriptive metadata enhances the model's understanding of the contextual use of each space, fostering a more accurate semantic interpretation of generated outputs. Additionally, this dataset presented a huge advantage compared to other tested datasets, the number of rows. Other explored and tested datasets presented a few hundred and at most a thousand rows, making it hard to train reliable and accurate models.

### 2.2 ADVANTAGES AND DISADVANTAGES OF USING THE DATASET

**Advantages**

- **Number of Observations:** The diverse number of images makes the dataset robust enough so we can train a model using the 12K images.
- **Resolution:** The resolution of 512 x 512 pixels ensures that the model has enough detail to generate precise and usable floor plans without consuming too much memory.
- **Annotations:** The inclusion of plain English descriptions provides an additional semantic layer that can be used in future improvements.

**Disadvantages**

- **Inconsistency in Annotations:** Variability in the quality and detail of the plain English descriptions can affect the consistency of the learning process in applications that combine language.

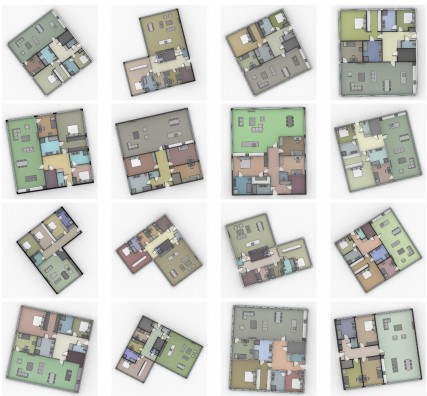

Figure 1: Selection of 16 random floor plans from the dataset

## 3 DIFFUSION MODELS

### 3.1 OVERVIEW OF DIFFUSION MODELS

Diffusion models are a class of generative models that have gained prominence for their ability to produce high-quality, high-fidelity images. Essentially, diffusion models work by gradually transforming a distribution of random noise into a structured image over a series of iterative steps, each guided by a learned reverse diffusion process. This process is often visualized as reversing the diffusion of particles from a concentrated point outwards, hence the name "diffusion models" Sohl-Dickstein et al. (2015).

## 3.2 RATIONALE FOR USING DIFFUSION MODELS

The decision to utilize diffusion models in the generation of architectural floor plans was driven by several factors. First, the nature of diffusion models to generate images through a gradual refining process allows for the capture of intricate details that are essential in architectural designs, such as the layout of rooms, placement of doors and windows, and the delineation of living spaces. This ability to manage and manipulate fine details aligns well with the requirements for generating usable and precise architectural plans. Furthermore, diffusion models have demonstrated a lower tendency towards mode collapse compared to GANs, offering a richer diversity in generated outputs which is beneficial for exploring a wide range of architectural styles and configurations Ho et al. (2020).

## 3.3 CURRENT STATE OF DIFFUSION MODELS

Recent advancements in diffusion models have established them as a leading technology in image generation. The introduction of conditional diffusion models has further expanded their applicability, allowing for the generation of images based on specific conditions or attributes, thereby increasing their utility in tasks requiring high degrees of specificity and customization Dhariwal & Nichol (2021). In the field of architectural design, these advancements present a promising avenue for not only generating floor plans from scratch but also modifying existing ones according to precise user specifications. The state-of-the-art models now feature improvements in training efficiency and the quality of the generated images, which make them particularly suitable for applications where detail and accuracy are paramount Nichol & Dhariwal (2021).

# 4 ARCHITECTURE OF THE U-NET MODEL

## 4.1 HISTORY AND BACKGROUND OF THE U-NET

The U-Net model was originally developed for biomedical image segmentation. The model was designed to work with a limited amount of data but still deliver strong performance, which was crucial in the medical imaging field where high-quality, annotated datasets are often small. U-Net's architecture is particularly noted for its effectiveness in handling multi-class segmentation Ronneberger et al. (2015).

## 4.2 DESCRIPTION OF THE ARCHITECTURE

The architecture of U-Net is characterized by its symmetric shape, which gives it the name "U-Net." It consists of a contracting path to capture context and a symmetrically expanding path that enables precise localization. The model uses a series of convolutional layers and max pooling layers in the contracting step to extract features and reduce the spatial dimensions of the input image. Each step in the expansive path consists of an upsampling of the feature map followed by a convolutional layer that halves the number of feature channels. A crucial feature of U-Net is the concatenation of feature maps from the contracting path to the upsampled output, a method known as skip connections. These connections help the network to propagate context information to higher resolution layers, allowing for more precise localization Ronneberger et al. (2015).

## 4.3 DIFFERENCES BETWEEN U-NET MODELS AND OTHER MODELS

Unlike many deep learning models that primarily focus on down-sampling to learn increasingly abstract representations, U-Net maintains a large amount of high-resolution information through its expansive path. This is in contrast to models like the standard convolutional networks, which may lose important local information due to repeated pooling operations. The ability of U-Net to maintain high-resolution details makes it exceptionally good at capturing the nuances of images, which is essential for tasks requiring precise segmentation and detailed reconstructions.

## 4.4 SUITABILITY OF U-NETS FOR ARCHITECTURAL FLOOR PLAN GENERATION

U-Nets are particularly well-suited for the task of generating architectural floor plans due to their powerful segmentation capabilities and the ability to handle fine-grained details—a necessity in

architectural designs where accuracy in the layout and clear differentiation of spaces are crucial. For this research, the U-Net's ability to effectively process and reconstruct complex spatial relationships within an image allows it to generate detailed and precise architectural layouts from a high-dimensional latent space. Furthermore, the skip connections in U-Nets help in recovering the exact spatial hierarchies and relationships of different architectural elements, such as walls, doors, and room designations, ensuring that the generated floor plans are not only aesthetically pleasing but also architecturally coherent.

## 5 DATA PREPARATION

### 5.1 OVERVIEW OF DATA PREPARATION

Early iterations of the model, which utilized smaller datasets, demonstrated limited variability in results, with the model failing to accurately interpret the intricacies of floorplans. For example, there were instances where rooms were depicted without proper access. Similar issues were observed when testing the model with a plain, untransformed dataset, resulting in images of suboptimal quality. Consequently, to enhance the generation of high-quality architectural floor plans using diffusion models, we implemented a rigorous preprocessing routine. This process involved standardizing and optimizing the dataset to better suit the specific requirements of this application.

### 5.2 TRANSFORMATION OF THE ENTIRE DATASET

The dataset transformation process involves several key steps designed to ensure that the input images are in a uniform format, which helps in reducing model complexity and improving training efficiency and most importantly, the final generated images. Here is an outline of the transformations applied:

**a) Detection of the Floor Plan from the Image**

- **Objective:** Isolate the architectural floor plan from any extraneous elements within the image. More specifically, separate the floor plan from the colored background.
- **Method:** Implement a detection algorithm that utilizes the wall outline image attribute from the dataset to distinguish the walls of the floor plan from the colored background.

**b) Rotation to Align the Floor Plan**

- **Objective:** Given that the dataset presents floorplans with different orientations, here we want to ensure that all floor plans are oriented in the same direction, which is essential for consistent model training.
- **Method:**
  - **Contour Detection:** The process begins by converting the mask image, which highlights the floor plan's footprint, to grayscale using the `cv2.cvtColor` function from the OpenCV library Bradski (2000). This grayscale image is then binarized using the `cv2.threshold` function to create a binary mask where the floor plan is represented by white pixels, and the background is black.
  - **Edge Detection:** To refine the contours, the grayscale image is blurred using a Gaussian filter `cv2.GaussianBlur` to reduce noise, and then edges are detected using the Canny edge detection algorithm `cv2.Canny` Bradski (2000).
  - **Angle Calculation:** The largest contour, corresponding to the floor plan, is identified using the `cv2.findContours` function. From this contour, the minimum area rectangle that can enclose the floor plan is computed using `cv2.minAreaRect`. The angle of this rectangle is extracted, and a correction is applied if the angle is less than -45 degrees. This correction is necessary to ensure the correct orientation, as angles close to -90 degrees would otherwise flip the floor plan Bradski (2000).
  - **Rotation Application:** Once the correct rotation angle is determined, the floor plan is rotated using the `cv2.getRotationMatrix2D` function to create a rotation matrix, followed by `cv2.warpAffine` to apply this rotation. The

cv2.warpAffine function is configured with cv2.INTER_CUBIC interpolation for the image and cv2.INTER_NEAREST for the mask to preserve edge sharpness, ensuring the rotation does not introduce padding or distortions. The cv2.BORDER_REPLICATE mode is used for the image rotation to handle border pixels, while cv2.BORDER_CONSTANT with a zero border value is used for the mask Bradski (2000).

**c) Background Standardization**

- **Objective:** Homogenize the background of all images to a uniform color, enhancing the model's ability to focus on the structural elements of the floor plans.

- **Method:** The background of each image is standardized by first checking if the image contains an alpha channel (transparency) and converting it to a standard RGB format using OpenCV's cv2.cvtColor function Bradski (2000), if necessary. A binary mask is then created by thresholding the grayscale version of the mask, and inverting it so that the background is highlighted. This binary mask is expanded to match the RGB channels of the original image using cv2.cvtColor, ensuring compatibility for background replacement. The replacement is performed by creating a new background array filled with the target color (white) using NumPy Harris et al. (2020), and applying it to the non-floor plan areas indicated by the mask. This process is executed using a custom function, ensuring that all images have a consistent, uniform background color.

Figure 2 displays a batch of floor plan images after getting transformed.



Figure 2: Selection of 16 random floor plan images after getting transformed

## 6 TRAINING

### 6.1 TRAINING CONFIGURATION

The training of the model followed this predefined configuration to ensure consistency and reproducibility of the results. The key parameters used within the training were as follows:

- **Image Size:** Each image was resized to 128x128 pixels to balance detail and computational efficiency.

- **Batch Sizes:** We utilized a training and evaluation batch size of 16.

- **Learning Rate and Scheduler:** An initial learning rate of 1e-4 was employed, with a warm-up phase of 500 steps to gradually reach the target rate.

- **Precision Settings:** To enhance computational efficiency, we employed automatic mixed precision (fp16), which significantly accelerated the training process without compromising the quality of the results.

## 6.2 UNet2DModel Architecture

We utilized the UNet2DModel from the diffusers library von Platen et al. (2022). The configuration for our model architecture was defined as follows:

- **ResNet Layers:** Each block in the U-Net structure contains 2 ResNet layers. This setup was chosen to leverage the robustness of ResNet architectures in feature extraction and representation learning He et al. (2015).

- **Block Channel Outputs:** The output channels for the blocks were set as follows: 128, 128, 256, 256, 512, 512. This progressive increase in channel depth allows the network to capture increasingly abstract features at different levels of the network.

- **Downsampling Blocks:** The model incorporated a mix of regular and attention-enhanced ResNet downsampling blocks. The configuration included four DownBlock2D regular blocks and one AttnDownBlock2D block, the latter integrating spatial self-attention to improve the focus on relevant features.

- **Upsampling Blocks:** Similar to the downsampling pathway, the upsampling path used a combination of regular UpBlock2D blocks and an AttnUpBlock2D block. The inclusion of attention mechanisms in the upsampling path helps in restoring image details more effectively during the generation process.

# 7 Enhancing Results through Upscaling Techniques

## 7.1 Introduction to Upscaling

Upscaling, in the context of image processing, refers to the technique of increasing the resolution of an image to enhance its resolution and detail. This is achieved by interpolating additional pixels into the original image using various algorithms designed to predict and replicate the underlying patterns and textures.

## 7.2 Rationale for Using Upscaling

In the generation of architectural floor plans using AI, the resolution of output images directly impacts their usability. Higher-resolution images can provide more detail, making them more practical for real-world applications. However, generating high-resolution images directly from the model can be computationally intensive and inefficient. By using upscaling, we can produce lower-resolution images during the generation phase—which requires less computational power—and subsequently enhance their resolution and detail post-generation, achieving great results at a lower cost.

## 7.3 Application of Upscaling Techniques

Following the generation of the floor plan images by our model, each image is processed through the upscaling tool. This post-processing step refines the visual details of the plans, enhancing lines, borders, and textures.

## 7.4 Trade-offs and Advantages

Opting to upscale lower-resolution images rather than directly generating high-resolution outputs presents several advantages. Primarily, it allows for faster model training and less intensive use of computational resources during the generation phase. Training models on high-resolution images not only requires significantly more memory and processing power but also increases the complexity of the model, leading to longer training time.

The trade-off here involves balancing the initial quality of generated images with the effectiveness of the upscaling process. By choosing to generate images at a lower resolution, we accept a compromise on initial detail with the understanding that the subsequent upscaling will restore and enhance these details efficiently. This strategy has proven effective, enabling the production of high-quality architectural floor plans with a fraction of the computational cost associated with high-resolution image generation.

## 8 EVALUATION

The evaluation of the generated architectural floor plans was conducted through a straightforward, yet effective qualitative analysis. Given the visual and functional nature of architectural designs, our primary evaluation method involved a direct visual inspection of the generated images. This approach allowed the research team to assess the practical utility and aesthetic quality of the floor plans in a manner that closely mirrors the real-world evaluation processes used by architects.

### 8.1 CRITERIA FOR EVALUATION

- **Accuracy:** Each generated image was reviewed to ensure that it accurately represented viable architectural spaces. This included checking for the presence and correct placement of essential elements such as walls, doors, and windows.
- **Coherence:** The coherence of the overall layout was assessed to determine if the generated designs made logical sense from an architectural standpoint. For instance, the flow between rooms, the functionality of the space, and the appropriateness of the design for hypothetical real-world applications were considered.
- **Aesthetics:** The aesthetic appeal of the floor plans was also a crucial evaluation criterion, reflecting the model's ability to generate visually attractive designs.

## 9 RESULTS

The performance of our model in generating architectural floor plans is demonstrated through a comprehensive analysis of training dynamics and visual results from both the initial generation and subsequent enhancement stages. This section presents the quantitative and qualitative outcomes of our experiments.

### 9.1 INITIAL RESULTS AND SUBSEQUENT IMPROVEMENTS

The initial results served as a fundamental benchmark for the subsequent refinements and enhancements applied to the model. These early results were essential in identifying key areas for improvement and in setting the trajectory for the research.

Figure 3 shows the initial batch-generated images.

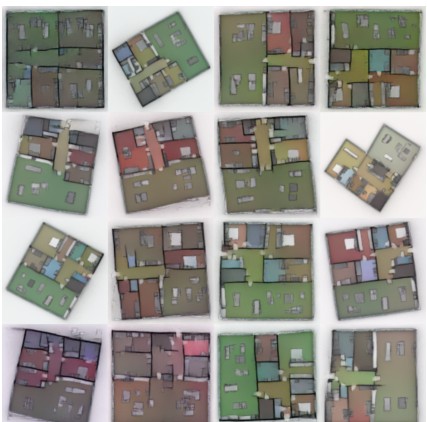

Figure 3: Initial batch of generated images

### 9.2 GENERATED FLOOR PLANS

After training, the model produced a series of floor plan images that were successfully evaluated for their quality and coherence. Figure 4 shows a batch of generated floor plans.



Figure 4: Batch of 16 generated floor plan images without enhancement

### 9.3 ENHANCED IMAGES THROUGH UPSCALING

To further enhance the utility and clarity of the generated images, an upscaling technique was applied post-generation. This step significantly improved the resolution and detail of the images, making them more practical for professional use. Figure 5 shows a batch of generated images with upscaling.

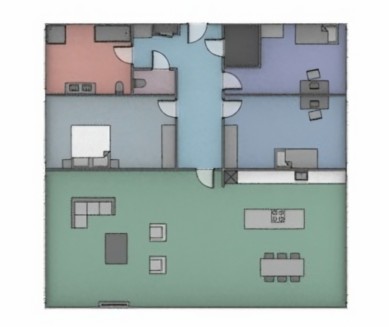

Figure 5: Batch of 16 generated floor plan images with upscaling

Figure 6: Close up of a single generated floor plan with enhanced quality

## 10 Applications

The advancements achieved in this project not only enhance the capabilities of generative models in producing architectural floor plans but also open up a multitude of practical applications across different sectors. Below are several key applications that demonstrate the wide-reaching impact and utility of this research.

### 10.1 Inspiration for Architects

The ability to generate diverse and innovative floor plans automatically provides architects with a powerful tool for inspiration. By generating a variety of design alternatives quickly, architects can explore creative possibilities without the usual time constraints associated with manual drafting. This can lead to the discovery of novel design solutions and architectural innovations, pushing the boundaries of traditional architectural practices.

### 10.2 Guidance in Remodeling and Space Optimization

Our model can play a crucial role in the remodeling of spaces by providing optimized floor plans that make the best use of available space. This is particularly valuable in urban environments where space efficiency is paramount. The model can suggest configurations that maximize utility and comfort, taking into account the specific dimensions and constraints of the existing structures.

### 10.3 Automated Compliance Checks

Integrating the model with tools that evaluate compliance with local building codes and regulations could significantly streamline the design and approval processes. By automatically generating designs that are pre-checked for compliance, the model can reduce the time and effort needed for regulatory approvals, making the construction process faster and more efficient.

## 11 Future work

The promising results achieved thus far in our project pave the way for several exciting directions in future research. Building upon the foundation laid by our current model, we aim to explore advanced techniques that could further enhance the flexibility, precision, and utility of our AI-driven architectural design tools.

### 11.1 Development of a Conditional Model

One of our primary objectives moving forward is the development of a conditional model that incorporates user-defined prompts to guide the generation process. This approach would allow users to specify certain characteristics or elements they wish to see in the floor plans, such as "a large bay window facing south" or "an open kitchen layout." By integrating natural language processing (NLP) techniques with our diffusion model, the system could interpret these prompts and directly incorporate the specified features into the generated designs.

### 11.2 Incorporation of Infilling Techniques

Another exciting avenue for future research involves the use of infilling techniques. Infilling allows for the selective generation of parts of an image, based on either predefined or dynamically chosen areas within a layout. In the context of architectural floor plans, this would enable users to select an area of an existing plan and request modifications or complete redesigns of just that section.

## 12 Related Work

The intersection of artificial intelligence with architectural design has garnered increasing research interest, particularly in the application of generative models and automation techniques to enhance

design efficiency and creativity. Generative Adversarial Networks (GANs) Goodfellow et al. (2014) and their adaptations, such as DCGANs Radford et al. (2015), have set benchmarks in realistic image generation across various domains, including architecture. Early efforts in architectural design automation, such as the automatic generation of building layouts Merrell et al. (2010), along with more recent developments like ArchGAN Oyelade & Ezugwu (2021), highlight the potential of GANs in transforming architectural visualization and planning. Moreover, the use of upscaling techniques, such as Super-Resolution Convolutional Neural Networks Dong et al. (2015), has significantly improved the quality of images, including those in architectural floor plans. However, historical attempts to automate floor plan generation using constraint-based algorithms Liggett & Mitchell (1981) and evolutionary algorithms Michalek et al. (2002) have faced challenges in flexibility, computational efficiency, and meeting user-specific aesthetic and functional requirements Knecht et al. (2010). Recent approaches, like those presented in House-GAN Nauata et al. (2021), attempt to address these issues but still struggle with ensuring that generated plans meet all practical and regulatory standards. Our research builds on these foundations, leveraging advances in diffusion models and upscaling techniques to enhance the automated generation of detailed and functional architectural floor plans, addressing longstanding challenges in computational efficiency and design adaptability.

## 13 CONCLUSIONS

This research set out to rigorously examine the applicability of state-of-the-art diffusion models tailored specifically to the task of generating architectural floor plans. By innovatively adapting these models to address the intricate requirements of detailed design tasks, our study has not only demonstrated the feasibility of such adaptations but has also propelled the capabilities of generative models to new heights. These findings affirm the potential of AI to master and implement complex 'implicit, human-learned' semantics necessary for practical and aesthetic architectural design.

### 13.1 ACHIEVEMENT OF SPECIFIC USE-CASE ADAPTATION

One of the primary motivations behind this work was to find whether SOTA models could be adapted to a specific use case that requires a high level of detail, such as architectural floor plans. Our findings affirmatively demonstrate that with careful configuration and training, diffusion models can indeed be specialized to handle such detailed tasks. The success in generating detailed, coherent, and structurally coherent floor plans validates our approach and underscores the adaptability of diffusion models to specialized domains.

### 13.2 LEARNING IMPLICIT, HUMAN-LEARNED SEMANTICS

A critical aspect of our research was to test if these models could learn and replicate 'implicit, human-learned' semantics necessary for practical architectural design, such as the logical placement of rooms and the structural necessities of buildings. The models not only learned these semantics but were also able to apply them creatively, suggesting new design possibilities that maintain both aesthetic appeal and functional integrity. This capability signifies a profound step forward in applying AI in design, bridging the gap between human expertise and machine-generated innovations.

### 13.3 COMPUTATIONAL EFFICIENCY THROUGH UPSCALING TECHNIQUES

The research further explored the efficacy of training smaller, less resource-intensive models combined with post-process upscaling techniques instead of relying on larger, more computationally demanding models. The results were clear: smaller models, when augmented with advanced upscaling technologies, could achieve comparable, if not superior, results in generating high-resolution outputs. This approach conserves computational resources and enhances the scalability and accessibility of AI technologies in architectural design, making it feasible for more firms to adopt this technology without the need for extensive hardware investments.

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
