# OpenReview forum: "From Abstract Noise to Architectural Form: Designing Diffusion Models for Efficient Floor Plan Generation"
_ICLR.cc/2025/Conference — Submitted to ICLR 2025_

### Official Review · Reviewer_pgZ4 · 2024-10-27

**Soundness:** 2
**Presentation:** 1
**Contribution:** 2
**Rating:** 3
**Confidence:** 5

**Summary:**

This paper proposes an application of diffusion models to the generation of architectural floor plans, fine-tuning diffusion models to learn implicit design concepts in architectural design, and generating detailed and functional architectural floor plans.

**Strengths:**

The paper explores the intersection of AI and architectural design, achieving promising visual results. I highly appreciate and commend the authors' attempt. However, it needs significant improvements for a top-tier conference.

**Weaknesses:**

**Lack of related work:** The paper proposes a new pipeline for generating floor plans. However, the authors lack a substantial amount of related work, including AI-assisted architectural design and generative model-related work.

**Limited technical innovation:** The technical innovation in this paper is quite insufficient, and the introduction to the U-Net architecture is entirely superfluous.

**Missing dataset:** It would be beneficial to introduce more types of architectural styles and layouts. The current dataset is still quite limited (Residential floor plan only ).

**Insufficient evaluation:** The paper lacks quantitative metrics and comparisons of related methods, including how to assess the rationality of generated floor plans. For design tasks, more professional architectural designers' user evaluations may be needed.

I consider this to be an inspiring report on the interdisciplinary area. I encourage the authors to conduct more detailed technical innovation and experimental evaluation. For ICLR, this paper clearly lacks innovation and systematic methodology. Therefore, I believe this paper would be more suitable for submission to architectural design-related conferences, as it does not quite meet the threshold for current AI conferences.

**Questions:**

Please see weaknesses.

---

### Official Review · Reviewer_Te5N · 2024-11-01

**Soundness:** 2
**Presentation:** 2
**Contribution:** 2
**Rating:** 3
**Confidence:** 3

**Summary:**

The paper is a report for training unet as a diffusion model to generate floor plan images

**Strengths:**

These steps are clear and techniques are correct.

**Weaknesses:**

This paper does not seems to propose a method. It is a report to describe an experiment.
It describes how to process images, how to build unet, how to train, how to write data augmentation codes, how to use postprocessing like upscale. But I do not think it has proposed some methodology technically.

**Questions:**

This doc is a good report for platforms like Kaggle or HuggingFace Model. But conferences like ICLR are more oriented to research that propose new methods or applications.

---

### Official Review · Reviewer_VSic · 2024-11-03

**Soundness:** 1
**Presentation:** 2
**Contribution:** 1
**Rating:** 3
**Confidence:** 3

**Summary:**

The paper proposes using a diffusion model to generate architectural floor plans. The paper shows the process of constructing a dataset and training details.

**Strengths:**

The authors experimented with using a diffusion model to generate floor plan designs and presented several results.

**Weaknesses:**

1. **Excessive Unnecessary Details**: The paper contains considerable redundancy, with numerous unnecessary details, such as the advantages and rationale for using diffusion models, specifics of the U-net architecture, and exact function names from OpenCV in the code. These details occupy a significant portion of the content (around 50%) but do not provide valuable insights.

2. **Lack of Novelty**: The paper does not demonstrate sufficient contribution or value in terms of model design, dataset construction, or performance presentation.

3. **Disorganized Structure**: With a total of 13 primary headings, the paper’s structure is confusing for readers, making it difficult to grasp the core content. Many sections could be merged to improve readability.

4. **Poor Performance Presentation and Analysis**: The paper lacks numerical results and provides insufficient visual examples to adequately showcase the model’s performance.

**Questions:**

As shown in Weaknesses

---

### Official Review · Reviewer_mfMP · 2024-11-04

**Soundness:** 1
**Presentation:** 2
**Contribution:** 1
**Rating:** 3
**Confidence:** 3

**Summary:**

This paper presented an application of diffusion models to the generation of architectural floor plan images. They presented details of data preprocessing and hyperparameters of training these generative models, some qualitative results, and potential applications.

**Strengths:**

This paper has some strengths:

- Detailed presentation of training details
- Carefully designed data preprocessing procedure for detection and alignment of floor plans
- Reasonable Generation Results for a difficult domain, given architectural designs needs to be coherent and have clear layouts

**Weaknesses:**

This paper has a number of significant weaknesses:

- **Lack of quantitative results and comparison to prior work**: There are a number of quantitative evaluation metrics available for evaluating image generation quality, such as Frechet Inception Distance (FID). Moreover, there is no comparison to prior work that performs architectural floor plan generation [1].

- **Lack of objective, expert evaluation for qualitative analysis**: Even with the evaluation criteria listed by the author(s) in Section 8.1, some of these evaluations would be significantly strengthened if conducted by real architects, or practitioner(s) with significant architectural experience. It is unclear if the research team has such expertise.

- **Limited Novelty of Application or use of Diffusion Models**: There are ample prior work for using Diffusion Models for Architectural Floor plan generation [1] or other kinds of layout generation [2], which reduces the novelty of this work. The author(s) also did not cite these other related work and/or discuss the relationship/difference between the presented work and prior work.

References:

[1] HouseDiffusion: Vector Floorplan Generation via a Diffusion Model With Discrete and Continuous Denoising. Mohammad Amin Shabani, Sepidehsadat Hosseini, Yasutaka Furukawa. CVPR 2023

[2] LayoutDM: Discrete Diffusion Model for Controllable Layout Generation. Naoto Inoue, Kotaro Kikuchi, Edgar Simo-Serra, Mayu Otani, Kota Yamaguchi. CVPR 2023

**Questions:**

- Does the research team have architecture expertise to conduct the qualitative evaluation in the paper?
- Can the author(s) provide more details about the architecture floor plan dataset? Will it be open-sourced?
- Can the author(s) provide quantitative results using established metrics, such as FID?

---

### Meta-Review · Area_Chair_G7Z5 · 2024-12-17

**Metareview:**

This research introduces a novel application of diffusion models adapted for the generation of architectural floor plans. The paper gives implementation and training details and demonstrates their method with several results. However, there are concerns over the technical contributions, novelty of the paper. Moreover, the proposed method is not well-evaluated. Therefore, I don't recommend this paper.

**Additional Comments On Reviewer Discussion:**

There is no discussion.

---

### Decision · Program_Chairs · 2025-01-22

Reject